# Characterisation of Cellulolytic Bacteria Isolated from Agricultural Soil in Central Lithuania

Arman Shamshitov [1,*], Francesca Decorosi [2], Carlo Viti [2], Flavio Fornasier [3], Gražina Kadžienė [4,*] and Skaidrė Supronienė [1]

[1] Microbiology Laboratory, Institute of Agriculture, Lithuanian Research Centre for Agriculture and Forestry, Instituto al. 1, Akademija, LT-58344 Kedainiai, Lithuania
[2] Genexpress Laboratory, Department of Agronomy, Food, Environmental and Forestry (DAGRI), University of Florence, Via Della Lastruccia 14, I-50019 Sesto Fiorentino, Italy
[3] Research Center for Viticulture and Oenology (CREA-VE), I-31015 Conegliano, Italy
[4] Department of Soil and Crop Management, Institute of Agriculture, Lithuanian Research Centre for Agriculture and Forestry, Instituto al. 1, Akademija, LT-58344 Kedainiai, Lithuania
* Correspondence: arman.shamshitov@lammc.lt (A.S.); grazina.kadziene@lammc.lt (G.K.); Tel.: +370-631-14-191 (A.S.); +370-686-49-431 (G.K.)

**Abstract:** Characterisation and evaluation of soil bacteria were conducted in order to select the most potent strains that participate in the degradation of cellulose in unique agroecosystem and climatic conditions. Cellulolytic activity of soil bacteria was estimated using qualitative assays such as growth on selective media followed by screening with Congo red, Gram's iodine solution, confirmation test on Congo red agar, determination of enzyme production, and sugar utilisation pattern. A total of 159 soil cellulolytic bacterial strains were selected based on shape, size, and colony characteristics. According to the results of all three screening assays, sixty-four, thirty-eight and fifty-one isolates were able to degrade at some level of cellulose, respectively. Partial sequencing of the 16S rRNA gene of 64 bacterial strains obtained using sequences retrieved from the databases indicated the presence of cellulolytic bacteria represented by members of the phyla *Actinobacteria* (48.44%), followed by *Firmicutes* (32.81%), *Proteobacteria* (15.62%) and *Bacteroidetes* (3.13%). Determination of enzyme production showed that fifteen strains possess endoglucanases activity which ranged from 9.09 to 942.41 nanomoles of MUF (4-methylumbelliferone) mL. Likewise, β-glucosidase enzyme activity was determined in 23.4 % of all isolates. The sugar utilisation pattern of soil bacterial strains displayed the different capabilities of growth and utilisation on various carbon sources, which occur in lignocellulosic materials (cellulose, starch) or their hydrolysates (glucose, galactose, fructose, cellobiose, maltose, lactose, sucrose, trehalose).

**Keywords:** crop residue; cellulolytic bacteria; cellulase production; 16S rRNA





## 1. Introduction

Soil conservation and restoration are the most significant issues in sustainable agriculture. In agricultural soils, crops determine the balance of organic matter synthesis, decomposition, and the organic waste they leave behind. Understanding the microbial decomposition of remaining or applied cereal crop residues in the soil is essential for soil management purposes such as preserving and replenishing soil organic matter [1]. Crop residues are materials left over after harvest in shapes like straws, roots, husks, stalks, and leaves and represent one of the most abundant raw materials on Earth [2,3]. According to García-Condado and others' assessment of annual crop residue production in the European Union is estimated at 419 million tonnes of dry matter per year (reference period 2011–2015) [4]. It should be mentioned that returning crop residues to the soil is a well-known and practical approach for increasing soil fertility, recycling nutrients, and avoiding the depletion of soil organic carbon that respectively has a positive impact on the soil's

physical, chemical, and biological characteristics [5]. Crop residues are mainly composed of a complex mixture of three natural carbohydrate biopolymers-cellulose (30–50%), hemicelluloses (20–40%), and lignin (10–30%), that are closely linked by physical and chemical forces [6]. The soil degradation of plant cell cellulose and other biopolymers is essential to the terrestrial carbon cycle [7,8]. It is worth noticing that cellulose is one of the most abundant biopolymers of plant materials on Earth; consequently, the degradation of this substance is a crucial step in plant residue decomposition.

Soil microbial communities are widely established to be essential regulators of carbon processes and the nutrient cycle in soil [9,10]. Moreover, they play a pivotal role in plant residue decomposition since microorganisms are the main producers of enzymes that participate in the degradation processes of plant cell wall polymers, including cellulose, hemicellulose and lignin in soils [8]. Cellulases are enzymes that cleave the β-1,4 bond in the cellulose chain during the breakdown process of this polymer. These enzymes are usually categorised as endoglucanase, cellobiohydrolases (cellobiose release from both ends of the chain) and β-glucosidases (cellobiose transformation to glucose monomers) [11]. Certainly, fungi are microorganisms that are widespread in the environment, especially in soil and play an essential role in the breakdown process of plant cellulose by producing a wide variety of cellulase enzymes [12,13]. Cellulose hydrolysing ability is common and widely recognised among the members of *Ascomycota* and *Basidiomycota* [14,15]. In addition to fungi, bacteria contribute significantly to soil cellulose degradation [7]. Bacteria are currently considered promising prospects for future strategies due to their broad functional diversity and versatility [16]. It was reported that the increase of bacterial biomass along with bacterial diversity was observed in the later stages of the plant residue decaying process, where recalcitrant compounds predominate [17]. A high tolerance range determines the versatility of bacteria in the environment to temperature (11–75 °C mesophilic and thermophilic, respectively), pH (5.5–9.0 for most non-extremophilic bacteria) and salinity (halophiles might grow in the conditions of more than 30% (*w/v*) total salts) [18–20]. In addition to the aforementioned characteristics and the ability of soil bacteria to break down cellulose under both aerobic and anaerobic soil conditions [21], this makes them potentially significant contributors to in situ degradation of soil cellulose.

The utilisation of microorganisms for accelerating in situ residue decomposition can be one of the most environmental-friendly, economical and viable options for recycling crop residue in agriculture while at the same time improving soil properties. It is well known that species richness and the efficiency of soil microorganisms depend on prevailing environmental and climatic conditions. Therefore, this study aimed to characterise and evaluate potential bacterial strains isolated from agricultural soil in central Lithuania to degrade cellulose that highly contributes to the decomposition of crop residues. Pure culture isolation, culturing, and enzyme assays were used to identify and phenotype potential bacteria, screen them, and select representative soil-cellulolytic bacterial strains.

## 2. Materials and Methods

### 2.1. Soil Sample Collection

Soil samples for isolation of bacterial isolates were collected from two factorial field experiments conducted in the Institute of Agriculture, Lithuanian Research Centre for Agriculture and Forestry, where A factor is different tillage treatments, and B factor a cover crop management. This experiment was established in Akademija, Central Lithuania (55°23′50″ N, 23°51′40″ E) in 1956. Tillage treatments included ploughing (1G), harrowing (3G), and no-tillage (5G). All tillage treatments were with cover crop (T) and without a cover crop. The field experiment is arranged as a block design in four replications. Cereal cropping sequences consisting of five-member crop rotations: winter wheat (*Triticum aestivum* L.)-winter rape (*Brassica napus*)-spring wheat (*Triticum aestivum* L.)-spring barley (*Hordeum vulgare*)-pea (*Pisum sativum*) has a history of more than three complete rotation cycles. All fields are maintained according to common agricultural practices, including pesticide use. In 2020, the year of soil sampling, field pea was grown. Soil samples were collected

randomly from the superficial soil (0–10 cm), aseptically stored in sterilised plastic bags and transported to the laboratory at 4 °C. In the laboratory, the soil was sieved through a 2.5 mm sieve and stored at −20 °C until analysis was carried out. A total of four sub-samples per one field replication of each tillage treatment were collected and later homogenised, resulting in one composite sample for each treatment. Soil physical-chemical characteristics were determined using standard methods, and the results are shown in Table 1. Briefly, soil $pH_{KCl}$ was determined in 1 N KCl extract using the potentiometric method, plant available phosphorus and potassium content—according to Egner-RiehmDomingo (A-L) method, organic carbon content—by dry combustion method. Organic matter—according to EN 13039:1999, total nitrogen—according to EN 13342:2000 using a nitrogen distiller.

**Table 1.** Physico-chemical properties of experimental site.

| Experimental Side | Soil Characteristics | | | | | | |
|---|---|---|---|---|---|---|---|
| | Texture | $pH_{KCl}$ | $P_2O_5$, mg/kg | $K_2O$, mg/kg | Organic C, % | Total N, % | Organic Matter, % |
| 1G | Loam | 6.5 | 281 | 314 | 1.40 | 0.146 | 4.87 |
| 3G | Loam | 6.3 | 272 | 399 | 1.58 | 0.116 | 5.43 |
| 5G | Loam | 6.3 | 312 | 525 | 1.59 | 0.172 | 5.21 |
| 1TG | Loam | 6.6 | 365 | 401 | 1.40 | 0.148 | 5.36 |
| 3TG | Loam | 6.3 | 362 | 510 | 1.69 | 0.180 | 5.72 |
| 5TG | Loam | 6.3 | 336 | 489 | 1.49 | 0.174 | 5.41 |

*2.2. Isolation and Screening of Potential Cellulolytic Soil Bacteria*

Ten grams of each soil sample were diluted with 90 mL of sterile water. Then 1 mL of the soil suspension was diluted sequentially (tenfold) and used to isolate cellulolytic soil bacteria by the standard method pour plate technique in triplicate. Cellulolytic bacteria were isolated on two selective media: cellulose agar medium [cellulose 2.0 g/L, gelatine 2.0 g/L, $MgSO_4·7H_2O$ 0.25 g/L, $KH_2PO_4$ 0.5 g/L and agar 15 g/L] and carboxymethyl cellulose agar medium (CMC) [peptone 10.0 g/L, CMC 10.0 g/L, $K_2HPO_4$ 2.0 g/L, $MgSO_4·7H_2O$ 0.3 g/L, $(NH_4)_2SO_4$ 2.5 g/L, gelatine 2.0 g/L and agar 15 g/L] at 30 °C for 2 to 3 days in order to allow for bacterial growth. All different bacterial colonies that appeared on the plates of the two selective media were subcultured to obtain pure strains and subsequently screened for cellulase production.

Each isolate was individually streaked on the plates of CMC agar and incubated at 30 °C for five days. Two different methods detected the hydrolysis zone around the colonies: (i) after incubation, CMC agar plates were flooded with 1% (*w/v*) Congo red reagent [22] for 15 min at room temperature and then washed with 1 M NaCl; (ii) CMC agar plates were flooded with Gram's iodine solution (2.0 g KI and 1.0 g iodine in 300 mL distilled water) for 3 to 5 min [23]. The hydrolysis capacity values of primary and secondary screening were calculated as a ratio of clear zone size to colony diameter. The clear zone around the bacterial colonies indicated cellulose degradation.

$$\text{Hydrolysis capacity} = (\text{clear zone diameter})/(\text{colony diameter}), \qquad (1)$$

The cellulose-degrading ability of bacterial strains was confirmed by streaking on the cellulose Congo red agar medium with the following composition (g/L): $KH_2PO_4$ 0.5, $MgSO_4$ 0.25, cellulose 2.0, agar 15.0, Congo-Red 0.2, and gelatine 2.0, pH 6.8–7.2. Colonies showing a discolouration of Congo red were valued able to degrade cellulose [24]. Selected bacterial isolates were also tested for degradation of other main plant cell polymers, such as lignin and hemicellulose. To evaluate the ability of bacterial isolates to degrade hemicellulose colonies were spotted on Xylan agar medium with the following composition

(g/L): peptone 5.0, yeast extract 2.0, MgSO$_4$·7H$_2$O 0.5, NaCl 0.5, xylan 5.0, agar 20.0, pH 7.0. Further, the colonies were flooded with 1% (*w/v*) Congo red reagent, followed by distaining with 1M NaCl [25]. The hydrolysis capacity values of screening were calculated using the equation that was mentioned earlier. The isolates were further screened for producing lignin-degrading enzymes using methylene blue dye as an indicator [26]. Ligninolytic enzymes incur oxidation of methylene blue inducing discolouration. The isolated bacterial strains were streaked on Luria Bertani (LB) medium [peptone 10 g/L, yeast extract 5 g/L, sodium chloride 5 g/L, agar 15 g/L] added with methylene blue indicator dye (0.25 g/L). The plates were incubated at 30 °C for 72 h. Colonies that showed discolouration of the methylene blue dyes are considered positive lignin-degrading soil bacterial strains. Morphological characteristics of bacterial isolates, such as Gram staining and microscopic observation, were performed using standard methods in microbiology.

### 2.3. Molecular Identification of Bacterial Isolates

DNA samples were extracted from fresh bacterial culture using Quick-DNA™ Fecal/Soil Microbe Miniprep Kit from Zymo Research Corporation. The 16S rDNA was amplified with bacterial universal primer pair 27F 5'-(AGAGTTTGATCMTGGCTCAG)-3 and 1387R 5'-(GGGCGGWGTGTACAAG GC)-3' [27]. PCR reactions were performed in a total volume of 25 μL using of DreamTaq Green PCR Master Mix (2X) 12.5 μL, DNA free water 7.6 μL, 0.1 mg/mL of bovine serum albumin 2.5 μL, 0.2 μL of 10 μM each primer, and 2 μL of isolated DNA. (Thermo Scientific, Waltham, MA, USA). The PCR amplification was carried out in a PCR thermal cycler (Bio-Rad My cycler, Berkeley, CA, USA) using a hot-start procedure at 94 °C for 5 min. Conditions consisted of denaturation 35 cycles at 94 °C for 30 s, annealing at 55 °C for 30 s, and extension at 72 °C for 90 s, followed by a final extension step of 72 °C for 7 min. The PCR amplicons were purified with the Gene JET PCR Purification Kit (Thermo Scientific) and subsequently sequenced by Applied Biosystems 3730XL DNA Analyzer using the same primer set used in PCR amplification.

The sequences were processed and analysed using FinchTV V 1.4.0 and assembled using EMBOSS merger. Further, the sequences were compared with those in the GenBank database using the Nucleotide BLAST search program the National Center for Biotechnology Information (http://www.ncbi.nlm.nih.gov/ (accessed on 10 May 2022)). The most similar sequences for each bacterial strain and type strain sequences were downloaded from the nucleotide database of the National Center for Biotechnology Information (NCBI). The sequences of bacterial isolates as well as downloaded sequences were aligned and trimmed using ClustalW multiple alignments in the BioEdit v.7.2 software [28]. The sequences were then analysed for the selection of the best DNA/protein substitution model in the MEGA 11.0 program. Models with the lowest BIC scores (Bayesian Information Criterion) and AICc value (Akaike Information Criterion, corrected) are considered to describe the substitution pattern the best [29]. TN93 + G + I for Actinobacteria, K2 + G + I for Firmicutes, and Proteobacteria with Bacteroidota K2 + G were selected. A phylogenetic tree was constructed using the maximum likelihood method with 1000 non-parametric bootstraps used as replicates in the MEGA 11.0 program (Molecular Evolutionary Genetics Analysis, Version 11.0) [30].

### 2.4. Determination of Enzymatic Activity

Enzyme activities were quantified in soil extracts [31]. Briefly, enzymes were extracted by heteromolecular exchange using a 3% lysozyme solution [32]. Hydrolytic activities were quantified in microplates using 4 methyl umbelliferyl and 7 amino 4 methylcoumarine fluorogenic conjugated substrates (Biosynth); peroxidase activity was determined using 10-acetyl-3,7-dihydroxyphenoxazine (Ampliflu red Merck cat. no. 90101). Endoglucanases, arylsulfatase, β-glucosidase, β-galactosidase, β-mannosidase, chitinase, xylosidase, alpha-arabinase, acid-phosphomoinoesterase, inositol-phosphatase (phytase), butyrate-esterase, nonanoate-esterase and peroxidase were determined in 200 mM MES (morpholineptansulfonic acid) solution at pH 5.8. The activities of leucine aminopeptidase,

arginine-aminopeptidase, serin-like protease, pyrophosphatase-phosphodiesterase, and phosphodiesterase were determined in 200 mM tris–HCl solution at pH 7.5. Alkaline phosphomonoesterase activity was determined in 200 mM tris–HCl solution at pH 9.0. Quantitation of microbial biomass was carried out as described by Fornasier et al. and expressed as the amount of double-strand DNA (dsDNA) [33]. TMEV software was used to produce a heatmap of the enzymatic activity profile of the bacterial isolates [34].

### 2.5. Phenotypic Characterisation of Bacterial Strains

Sugar utilisation phenotype was valued for the bacterial isolates. Eight different sugars: glucose, galactose, fructose, cellobiose, maltose, lactose, sucrose, trehalose, as well starch and carboxymethylcellulose were tested. Bacterial strains were grown on Plate count agar medium [Tryptone 1 g/L, yeast extract 1 g/L, glucose 0.2 g/L, agar 16 g/L] and incubated at 30 °C for 48 h. Then colonies were picked up with a sterile cotton swab and suspended in 1 mL 0.8% (*w/v*) NaCl solution. Cell density ($OD_{600}$) was measured by Biophotometer (Eppendorf) and subsequently adjusted to $OD_{600} = 1$. The sugar utilisation tests were performed in Sugar Test Medium (STM) [33.7 mM $Na_2HPO_4$, 22 mM $KH_2PO_4$, 8.55 mM NaCl, 9.35 mM $NH_4Cl$, 1 mM $MgSO_4$ $7H_2O$, 0.3 mM $CaCl_2$, 1 mg/L biotin, 1 mg/L Ca-Pantothenate, 1 mg/L Thiamine, Na2EDTA 0.134 mM, 0.031 mM $FeCl_3$ $6H_2O$, 6.2 μM ZnCl2, 0.76 μM $CuCl_2$ $2H_2O$, 0.42 μM $CoCl_2$ $2H_2O$, 0.081 μM $MnCl_2$ $4H_2O$, 1.6 μM]. For each strain, STM (11.9 mL) was added with 0.625 mL of the previously prepared bacterial suspension ($OD_{600} = 1$) (inoculated medium, $OD_{600} = 0.05$) or with 0.625 mL 0.8% (*w/v*) NaCl solution (not inoculated medium). Then 225 μL of the inoculated and not inoculated media were dispensed into the wells of a microplate previously added with 25 μL of the different sugar solutions (2% *w/v*). The final concentration of the sugar in the STM medium was 0.2% (*w/v*). Also, one test set up in STM medium without sugar, and one test set up in Plate count broth [tryptone 1 g/L, yeast extract 1 g/L, glucose 0.2 g/L], were included. Each test was performed in quadruplicate (*n* = 4). Microplates were incubated at 30 °C for 5 days, and then the $OD_{590}$ was read by a GDV programmable MPT Reader DV 990BV4 (Agilent Technologies). For each sugar, the bacterial growth was expressed as $OD = O_{Din} − O_{Dn}$, where $O_{Din}$ is the average $OD_{590}$ detected in the inoculated medium (*n* = 4), and $O_{Dn}$ is the average $OD_{590}$ detected in the not inoculated medium (*n* = 4). On the basis of the calculated OD, TMEV software was used to produce a heatmap of the sugar utilisation profile of the bacterial isolates [34].

### 2.6. Statistical Analysis

All the results of screening for cellulose and hemicellulose degradation reported are the means of three replications. The normality of the data was evaluated using the Shapiro-Wilk test. Parametric (ANOVA) and non-parametric test (Kruskal-Wallis) statistical tests were used, respectively, for normally distributed data and not normally distributed data. Post hoc tests to compare the bacterial growth on STM added with the ten different sugars with respect to growth on STM without sugar (negative control) were performed by Tukey's HSD and Dunnet test, respectively, for normally distributed data and not normally distributed data. Square root transformation applied for non-normally distributed data. The principal component analysis (PCA) was performed by using R version 4.2.1.

## 3. Results

### 3.1. Isolation and Screening of Potential Cellulolytic Soil Bacteria

One hundred fifty-nine bacterial strains were isolated from agricultural soils of fields with different tillage treatments and cover crop management based on their capability to grow on a media containing a recalcitrant carbon source (CMC or cellulose). The results of screening using the Congo red staining to detect the cellulolytic activity of isolates grown on CMC media revealed that approximately 40.25% of bacterial strains (64 out of 159) were able to degrade CMC and were selected for further analysis. Microscopic examination

of soil bacterial isolates revealed that sixteen strains were gram-negative, and the rest forty-eight were gram-positive.

Likewise, it was observed that most of them were rod, filamentous, and cocci shaped (Table S1). According to the screening with Congo red, hydrolysis capacity values ranged from 1.4 (strain 3TG.21) to 8.0 (strain 5TG.25.2) and among all bacterial isolates, sixteen showed the highest cellulase activity (HC more than 5.0). However, detecting the hydrolysis zone around colonies grown on CMC media by Gram's iodine solution revealed that only 38 out of 64 strains exhibited cellulolytic activity. Bacterial strain 3G.18 showed the highest hydrolysis capacity, equal to 6.7. Colony discolouration in the further screening of soil bacteria on Congo red agar medium indicated cellulose degradation. According to the test, 79.69% of strains (51 out of 64) were confirmed to produce cellulase.

Additionally, the abilities of the strains to degrade lignin and hemicellulose were investigated. Changes in the colour of the LB agar plates containing methylene blue from blue to clear were observed for 7.81% of bacterial strains (5 out of 64), which confirms their ability to degrade lignin. Likewise, fourteen bacterial isolates showed the capability to participate in the degradation process of hemicellulose. The hydrolysis capacity value ranged from 1.3 (strain 5TG.32) to 8.0 (strain 1G.49) (Table S1). The principal component analysis eigenvectors of screening data reveal that Gram's Iodine (0.32) has a relative contribution to PC 1 and (0.57) significant contributions to PC 2 (Figure 1).

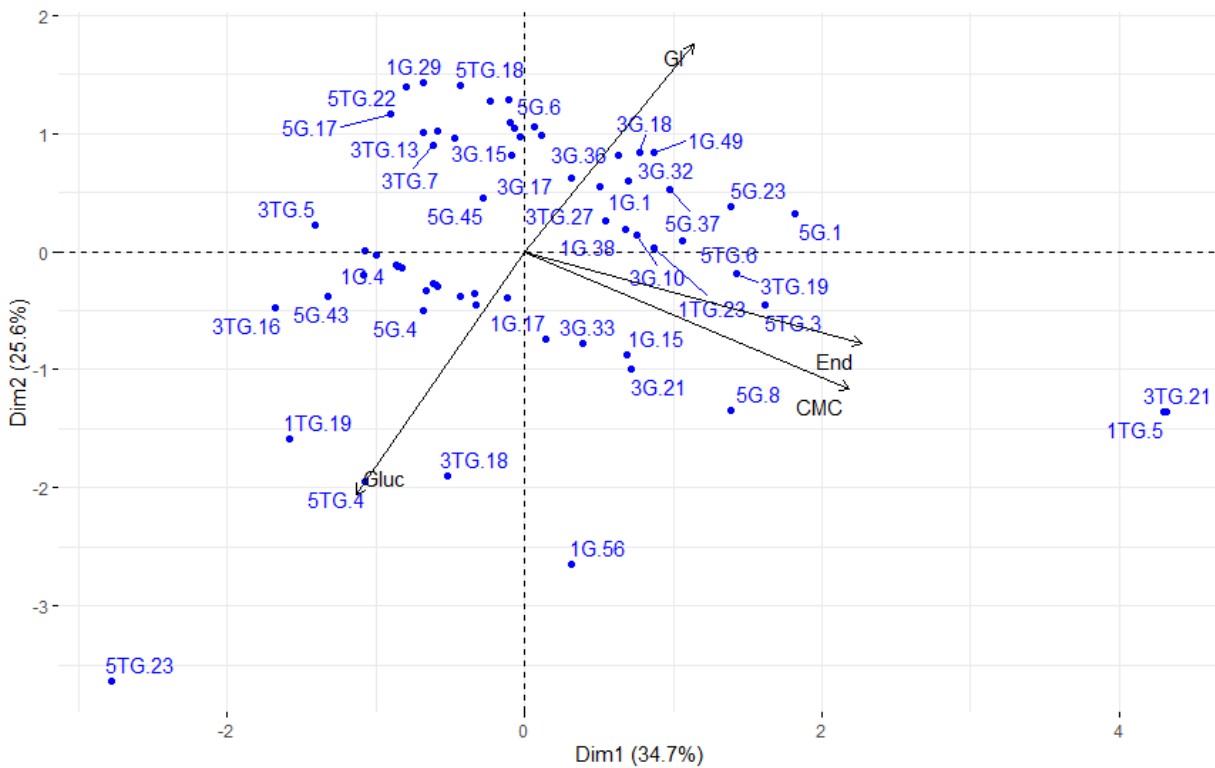

**Figure 1.** Principal component analysis of soil bacteria. GI: Gram's iodine, End: endoglucanase, Gluc: β-glucosidase, CMC: carboxymethylcellulose.

### 3.2. Identification of Bacterial Isolates

Sixty-four isolates were identified by 16S rRNA gene sequencing. The partial 16S rRNA sequences showed the highest similarity to sequences of the members of the phyla *Actinobacteria* (48.44%), followed by *Firmicutes* (32.81%), *Proteobacteria* (15.63%) and *Bacteroidetes* (3.13%). The 16S rRNA gene sequences were used to build phylogenetic trees to evaluate the phylogenetic relationships of the strains with respect to type strains of *Proteobacteria* and *Bacteroidota* phyla (Figure 2), *Firmicutes* phylum (Figure 3) and *Actinobacteria* phylum (Figure 4). All isolates were classified into 18 genera and 45 species. Among

all soil bacterial isolates, more than one strain of the following species *Stenotrophomonas rhizophila* sp., *Arthrobacter pascens* sp., *Paenarthrobacter nicotinovorans* sp., *Oerskovia paurometabola* sp., *Terrabacter carboxydivorans* sp., *Agromyces cerinus* sp., *Streptomyces canus* sp., *Streptomyces argenteolus* sp. *Bacillus pumilus* sp., *Bacillus altitudinis* sp., *Bacillus mobilis* sp, *Bacillus butanolivorans* sp. were identified.

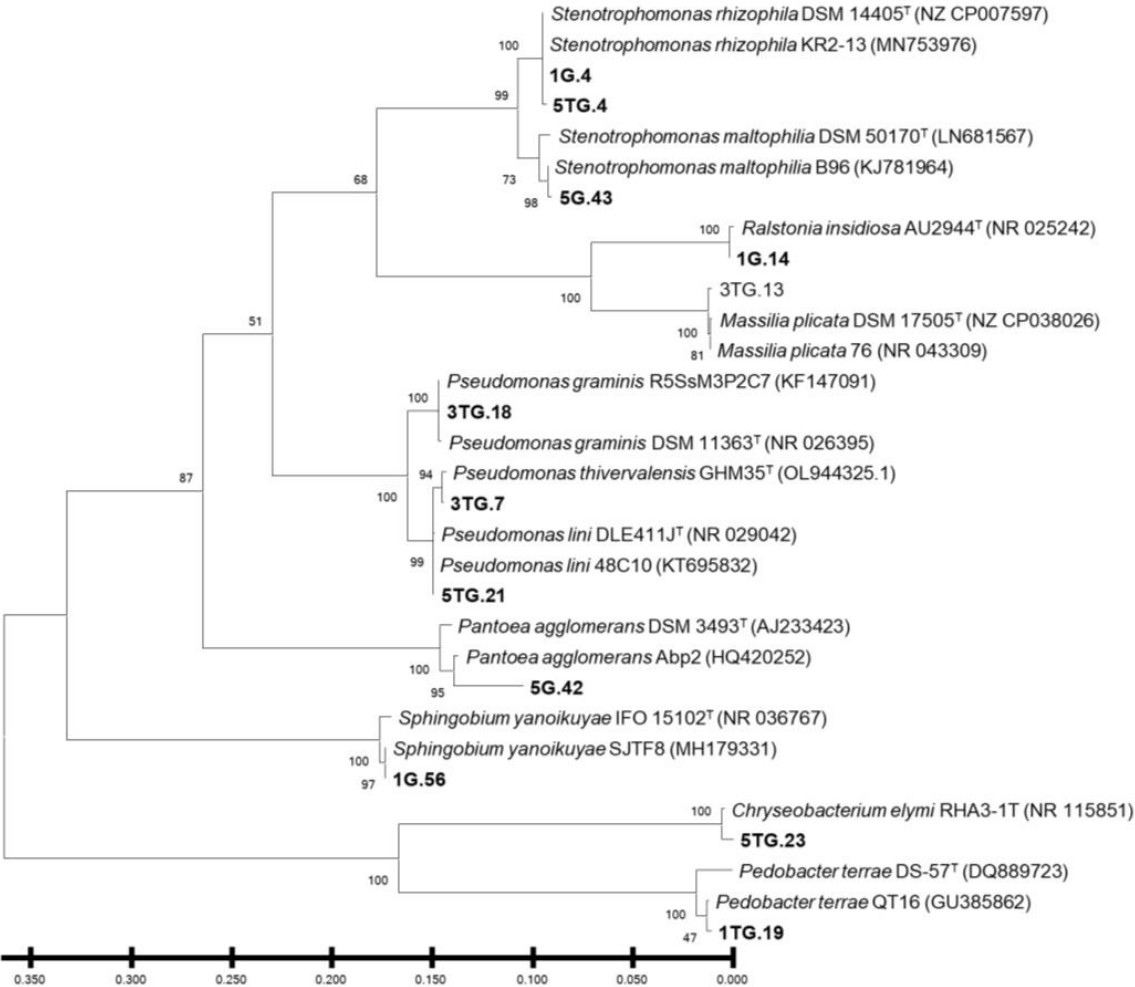

**Figure 2.** Phylogenetic tree of 16S rRNA gene of cellulolytic bacterial isolates that belong to *Proteobacteria* and *Bacteroidota* phylum.

The sequence data showed that eleven cellulolytic isolates were closely related (>98% similarity) to nine different species of the genus *Streptomyces* (Figure 4). Based on the results, we can consider that *Streptomyces* was the most dominant genus in the soil bacterial community. In this study, 21 bacterial strains belong to *Firmicutes* phyla which are represented by *Bacillus* and *Paenibacillus* genera, including twelve species (>98% similarity) (Figure 3). In accordance with the results, *Streptomyces* and *Bacillus* are the most widely distributed bacteria among all soil samples. Nine bacterial isolates belonging to the *Proteobacteria* phylum displayed high similarity (>99%) to the following genera *Stenotrophomonas*, *Ralstonia*, *Massilia*, *Pseudomonas*, *Pantoea*, *Sphingobium* (Figure 2). However, based on the BLAST search results, bacterial strain 5G.42 isolated from the soil sample No-tillage without cover crop exhibited comparatively low similarity (96.5%) to *Pantoea agglomerans* sp.

The strains 1G.4 and 5TG.4 isolated from soil samples ploughing without cover crop and No-tillage with cover crop showed 100% and 99.70% sequence identity to *Stenotrophomonas rhizophila* sp., respectively. Only two isolates belong to the *Bacteroidetes* phylum 1TG.19 and 5TG.23 isolated from the soil samples ploughing with a cover crop and No-tillage with a cover crop, respectively (99.39% and 99.15% identity). All the soil

bacterial strains were classified as bacteria in biosafety level 1, apart from 5G.42 and 5G.43 (level 2).

### 3.3. Determination of Enzymatic Activity

Extracellular enzyme secretion is essential in the degradation process of cellulose in the soil during the depolymerisation of crop residue. Presently, 44 families out of 115 O-glycoside hydrolases have been shown to contribute to plant cell wall breakdown. Therefore, various types of hydrolytic and one redox enzymes were assayed in cultures in order to get their enzyme profile. Bacterial strains showed a different pattern of enzyme activity (Figure 5). Endo-1,3(4)-β-glucanase (EC 3.2.1.4) hydrolyse internal glycosidic linkages in cellulose; in turn, cellobiose and other soluble glucooligomers are then converted to glucose by βeta-glucosidase (EC 3.2.1.21) [35].

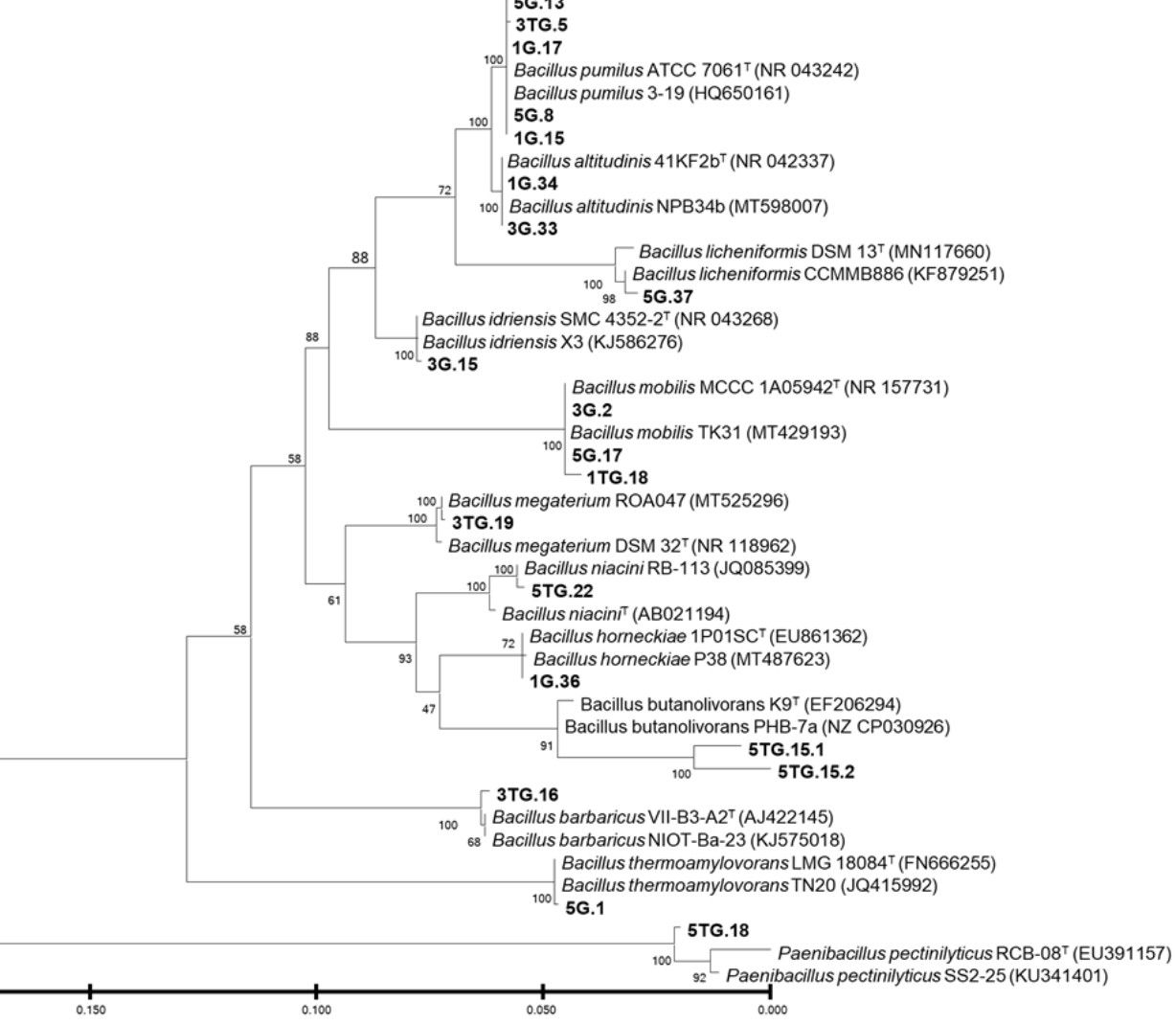

**Figure 3.** Phylogenetic tree of 16S rRNA gene of cellulolytic bacterial isolates that belong to *Firmicutes* phylum.

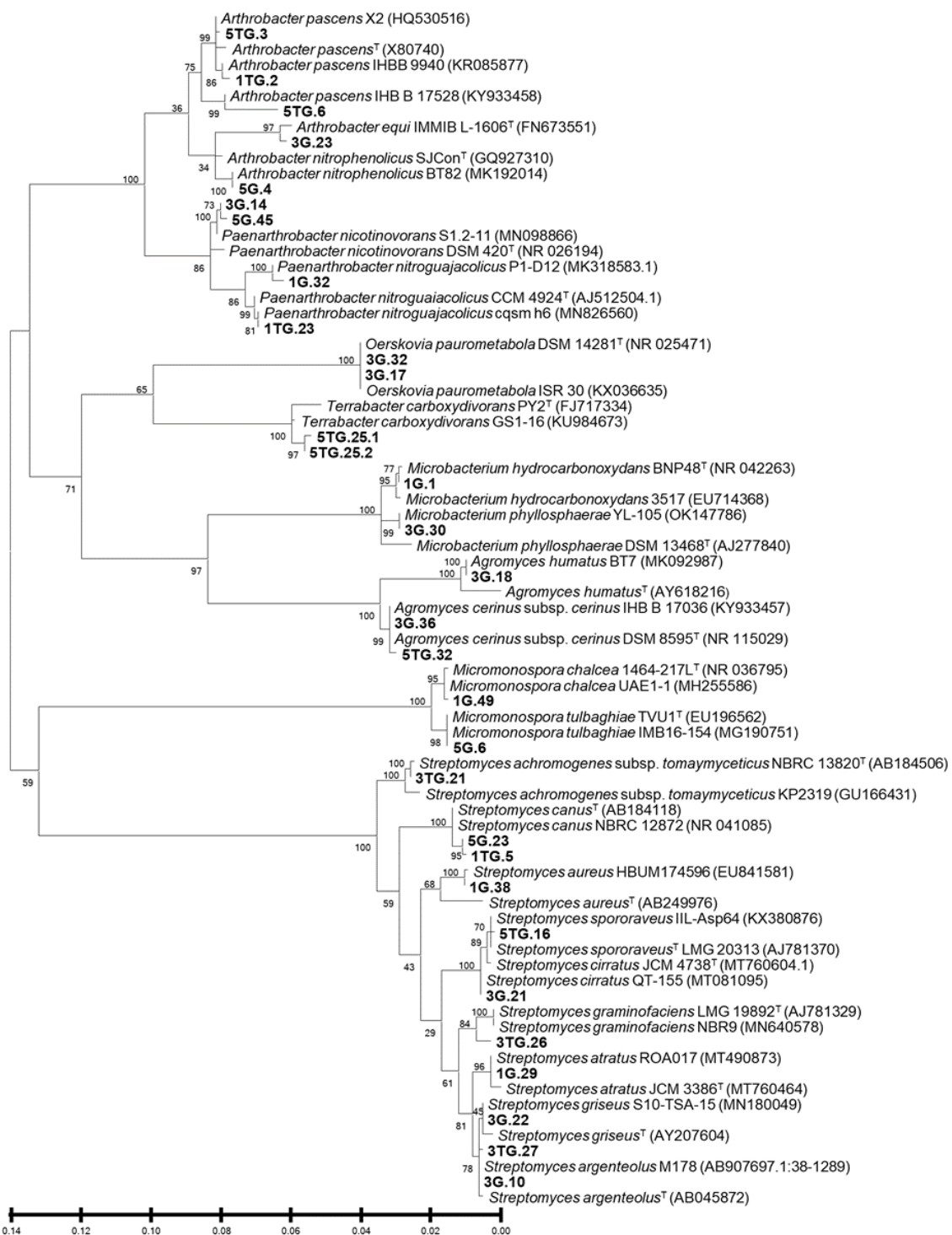

**Figure 4.** Phylogenetic tree of 16S rRNA gene of cellulolytic bacterial isolates that belong to *Actinobacteria* phylum.

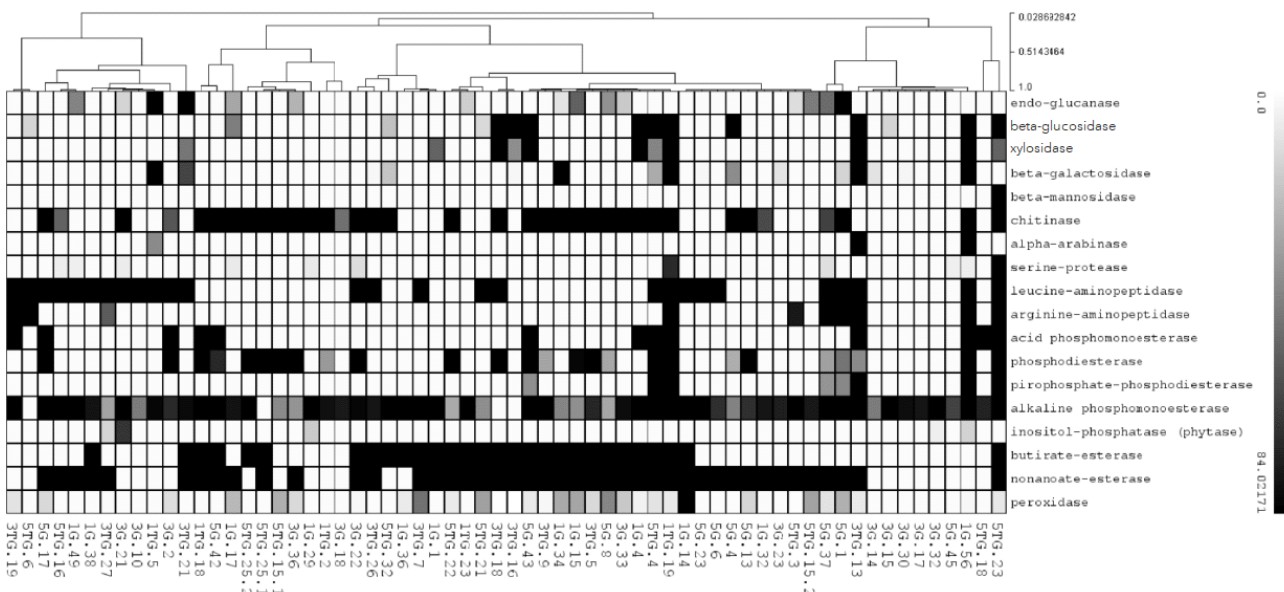

**Figure 5.** Heat map showing enzymatic activity of bacterial isolates. Enzymatic activity values are reported on a grey scale [White corresponds to 0.0 Black corresponds to 84.02171]. Data are expressed as an average activity.

Only following fifteen bacterial strains 1G.17, 1G.34, 1G.49, 1G.15, 3G.21, 3G.36, 3G.33, 5G.1, 5G.37, 5G.8, 1TG.5, 1TG.23, 3TG.21, 5TG.3, and 5TG.15.2 out sixty-four possess endoglucanases activity which ranged 9.09 to 942.41 nanomoles of MUF (4-methylumbelliferone) mL. β-glucosidase enzyme determined in 23.4% among all bacterial isolates, strains 5TG.23, 1G.56, 3TG.18 and 3TG.16 showed strong activity of this enzyme 40,298.40, 16,997.17, 13,191.29 and 12,766.87 MUF mL, respectively.

Our results show that only one strain 1G.17 (*Bacillus pumilus*) was positive for the two enzymes mentioned above, which might be explained by the use of resulting compounds in their metabolism rather than the simple sugars that would occur from entire cellulose breakdown. The strain 5TG.23 of *Chryseobacterium* genus expressed the highest β-glucosidase activity level and also was positive for xylosidase production (51.64 MUF mL). Xylosidases which participate in the degradation of hemicellulose with another broad set of enzymes were detected in 17% of strains. Strains 1TG.19 (*Pedobacter terrae*) and 5G.43 (*Stenotrophomonas maltophilia*) exhibited the highest activity of this enzyme 806.73 and 762.82 MUF mL, respectively.

Likewise, it is worth noting that the results of enzymatic activity assays demonstrate quite interesting phytase and phosphatase protein patterns, which differ among all bacterial strains. The PCA eigenvalues indicate that endoglucanase activity (0.64) significantly contributes to PC 1 (Figure 1).

### 3.4. Phenotypic Characterisation of Bacterial Strains

The growth of bacterial strains on different carbon sources, which occur in lignocellulosic materials (cellulose, starch) or their hydrolysates (glucose, galactose, fructose, cellobiose, maltose, lactose, sucrose, trehalose), was studied in order to determine and comprehend the metabolic versatility of soil bacterial isolates. The utilisation of different sugars, starch and CMC was investigated and shown in Figure 6 as a heat map which indicates the sugar utilisation pattern of bacterial strains. The heat map was created to better visualise the similarities and differences in the sugar utilisation pattern of soil bacterial strains. It was observed that some sugar sources were utilised differently and some similarly by bacterial isolates inhabiting agricultural soil. The highest rate of different sugars consumption was recorded for the following bacterial strains 1TG.5 (*Streptomyces*

*canus* sp.), 1TG.19 (*Pedobacter terrae* sp.), 5TG.25.1 (*Terrabacter carboxydivorans* sp.), and 5TG.22 (*Bacillus niacin* sp.).

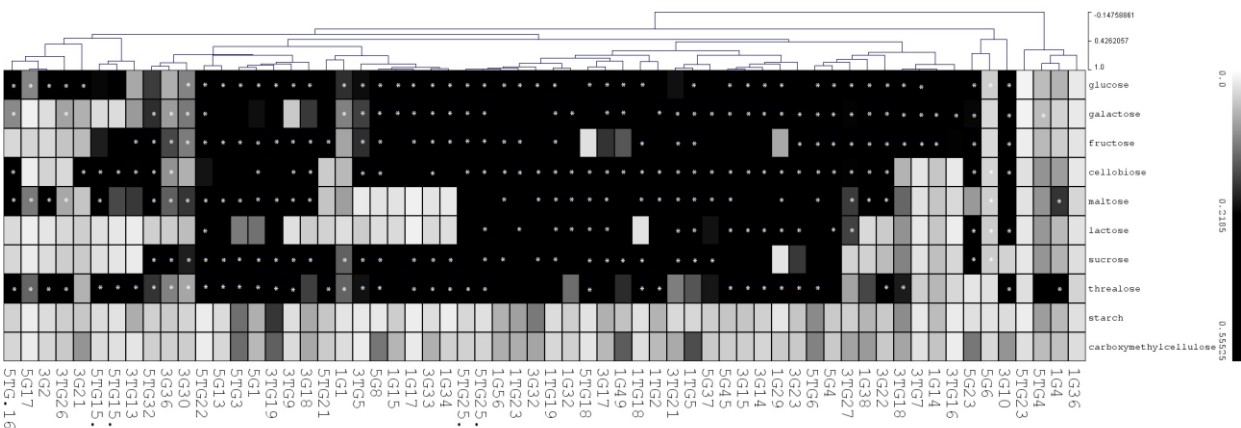

**Figure 6.** Heat map showing sugar test utilisation by the bacterial isolates. Data are expressed as average OD, calculated as reported in the text after five days of incubation at 30 °C ($n = 4$). OD values are reported in grey scale [White corresponds to OD = 0.0, black corresponds to OD = 0.55525]. Significant increases ($p < 0.05$) in the OD of the cultures grown on specific sugars with respect to cultures grown without sugars are marked by an asterisk.

Likewise, the most frequently utilised sugar sources were glucose, followed by galactose, cellobiose and trehalose. These sugars were consumed significantly with respect to cultures grown without sugar sources by 45, 40, 41 and 38 bacterial strains, respectively.

As expected, the heat map shows that with regard to other sugar sources, starch and CMC were used less, which the recalcitrant properties of these polymeric carbohydrates can interpret. Nevertheless, some bacterial strains showed high OD values of consumption of starch 0.1720 (3TG.19), 0.12275 (5TG.3) and CMC 0.1525 (1TG.5), 0.1390 (1G.49), respectively. The isolates 5TG.23 and 1G.36 were characterised by the lowest utilisation of all carbohydrate compounds. According to the heat map, in some cases, strains belonging to the same genera or species had similar patterns and were clustered in the same or close groups. For instance, the isolates 1TG.5 and 3TG.21 that belong to the *Streptomyces* genera utilised all the sugar sources correspondingly and were clustered into the same group. It was also observed that the following bacterial strains, 1G.4 and 5TG.4, belonging to *Stenotrophomonas rhizophila* sp. were clustered into the same group with comparatively similar lower abilities to induce sugar and polymeric carbohydrate sources. However, strains 1G.14, 3TG.7, and 3TG.16 belong to different taxa *Ralstonia insidiosa* sp., *Pseudomonas fluorescens* sp., and *Fictibacillus barbaricus* sp., respectively, also exhibited relatively similar sugar utilisation profile and accordingly were clustered into the same group. The PCA eigenvalues indicate that CMC utilisation (0.62) significantly contributes to PC 1 (Figure 1).

The total variance in data obtained from cellulolytic activity evaluation of bacterial isolates indicated by the first principal component axes was 60.3% (34.7% PC1 and 25.6% PC2) (Figure 1). Correlation analysis between original variables and PCs indicates a moderate positive correlation between PC 1 and GI (0.38), a very strong positive with endoglucanases (0.76), and a moderate-negative correlation with β-glucosidase (−0.38), a very strong positive with CMC utilisation (0.73) respectively. With respect to PC 2 and original variables, a strong positive correlation indicates with Gram's Iodine (0.58), a strong negative with β-glucosidase (−0.68), a weak negative correlation effect with endoglucanases (−0.26), and moderate-negative with CMC utilisation, respectively.

## 4. Discussion

Even though numerous bacterial strains from various soils have previously been isolated and identified, these studies are heavily biased towards a limited collection of

taxa from particular ecosystems. This study evaluated members of 18 genera isolated from agricultural soils in the Nemoral environmental zone conditions to degrade cellulose. The results of the screening assay using Congo red are consistent with previous studies [36,37], where it was observed the range of hydrolysis capacity values of 0.34–13.11. Kakkar et al. reported that Gram's Iodine has the greatest efficiency compared to Congo red and other dyes [38], which might explain the lower number of bacterial isolates that showed cellulolytic activity in this screening assay. Likewise, it should be noted that the formation of a halo zone depends on binding the dye to the degraded polymer as well as on the bacterial isolate. Among sixty-four isolates that showed cellulolytic activity at some level, twenty-five were positive in all screening assays, including Congo red, Gram's Iodine, and confirmation test on cellulose Congo red agar. Although many studies have used screening for cellulose degradation using Congo red agar and other dyes as a marker. However, some researchers report that hydrolysis capacity value may not accurately reflect the ability to produce cellulose-degrading enzymes [39,40]. Microbial lignin breakdown is essential in completing the carbon cycle since reducing the recalcitrant lignin barrier allows relevant microorganisms to access other plant cell polymers [41,42]. Capabilities of soil bacterial strains to break down and assimilate lignin as the primary carbon source aligned with Umashankar et al. findings [43].

The results of partial 16S rRNA sequences of sixty-four strains are in agreement with earlier reported studies where the most potential bacteria that carry cellulase genes are commonly associated with specific genera within *Actinobacteria*, *Firmicutes*, *Proteobacteria* and *Bacteroidetes* phyla [8,44,45]. 31 out of 64 strains were identified as bacteria belonging to different genera of *Actinobacteria* phylum. Previous studies confirm that *Actinobacteria* are one of the most widely distributed phyla among soil bacterial communities producing a wide range of enzymes that participate in plant residue decomposition [46–48]. It is worth noting that the *Streptomyces* genus of *Actinobacteria* phylum is essential due to their biological and functional activity, particularly in the soil environment where they ranked as vital contributors to the decomposition of cellulose and other plant cell biopolymers [49–51]. *Streptomyces achromogenes* sp., which wasn't previously mentioned as a cellulase-producing bacteria, exhibited comparatively the high endoglucanase activity (942.4 MUF mL). Likewise, numerous bacterial strains of *Bacillus* and *Paenibacillus* genera possess the ability to produce complex enzymes to degrade plant cell biopolymers, including cellulose, hemicellulose, and lignin, have been isolated from various soils worldwide [52–57]. Jain and others reported that inoculation with the CDB-16 strain of *Stenotrophomonas rhizophila* sp. of vegetable waste led to a weight loss of 30% and showed one of the highest cellulase enzyme activity among the other bacterial inoculants [58]. However, in our investigation, bacterial strain 1G.14 (*Stenotrophomonas rhizophila* sp.) showed positive results only for screening assay using Congo red and for beta-glucosidase production.

Cellulases collaborate with some other hydrolytic enzymes to accomplish the degradation of the polysaccharide to soluble sugars, specifically glucose and cellobiose, which are subsequently assimilated by the plant cell wall [59,60]. According to Morais et al., the combination of cellulose-degrading enzymes acts synergistically, causing complete external cellulose breakdown and perhaps a nutritional advantage for the producing organism [61]. Determination of cellulase–xylanase activity in strain 5TG.23 agrees with earlier reported studies where the first bifunctional cellulase–xylanase CbGH5 protein in *Chryseobacterium* genus was identified. In that study, the strain (*Chryseobacterium* sp. HT1) was isolated from cattle fed with cereal straw as the main carbon source [62]. Bacterial strain 1G.15 belonging to *Bacillus pumilus* sp. also exhibited endoglucanase activity (52.10 MUF mL). In a similar study, Chaudhary and others observed the occurrence of *Bacillus pumilus* sp., *Bacillus licheniformis* sp., *Paenibacillus dendritiformis* sp., and *Bacillus cereus* sp. from soil samples where sugarcane bagasse was grown; the strain NB-3 (*Bacillus pumilis* sp.) produced the highest amount of cellulase (13.6 5.6 mol/mL) when compared to other soil bacterial strains [63]. Strain 3TG.13, which belongs to *Massilia genus*, was positive for all qualitative cellulose degrading assays and showed beta-glucosidase activity (415 MUF mL).

With regard to studies on cellulolytic bacteria, Du and colleagues used complete genome research and found that the strain of the *Massilia genus* contains carbohydrate enzymes like glycoside hydrolase and polysaccharide lyase, allowing the strain to degrade cellulose [64]. In our investigation, four bacterial strains of the same species showed a different ability to produce cellulose-degrading enzymes. 1G.15 and 5G.8 showed only endoglucanase activity, whereas strains 5G.13 and 3TG.5 were negative for the production of both endoglucanase and beta-glucosidase enzymes. Possibly it can be explained by the individual ability of each strain to secret specific enzymes even if they represent the same species. Also, individual gene expression in a bacterial culture depends on the bacteria never being in the same growth phase and that it controls the expression of genes, including those encoding enzymes. According to enzymatic activity determination, it was also possible to verify that 35 of 64 soil bacterial isolates did not possess any cellulose-degrading enzymes. The results are comparable with the literature data, which also describes that most bacterial isolates with clearance zones in CMC agar had undetectable FPase activity [40].

In environmental conditions, microorganisms' survival can be predicted by the capability to utilise a range of available carbohydrates as carbon and energy sources [65]. The efficient uptake rate of glucose agrees with the results of Jame et al., which showed that all investigated bacterial isolates utilised this monosaccharide as a carbon source [66]. Due to the fact that glucose is the preferable sugar, most presumably because it can enter glycolysis directly, whereas other sugars need to be enzymatically transformed. According to the clustering, which found similarities in sugar utilisation patterns of soil bacterial isolates, separated them into multiple groups. The results, in most cases, agree with Suproniene and others' findings, where it was revealed that phenotypic differences in carbohydrate usage were mainly attributed to the taxonomic affiliations of the *Rhizobial* strains [67]. However, Mekonnen et al., who studied carbon substrate utilisation patterns of three ureolytic bacteria, concluded that even closely related species may have quite a distinct substrate usage patterns [68]. Therefore, a relationship between species occurrence and substrate turnover can not be assumed all the time.

## 5. Conclusions

Our study shows that by characterising culturable bacterial communities isolated from agricultural soil in the Nemoral environmental zone conditions, a variety of bacterial taxa that might be essential in the decomposition of cellulose in the soil was identified. Consequently, they may actively participate in the cycling of organic matter. Results also show that only the 1G.17 (*Bacillus pumilus* sp.) strain was positive to produce both endoglucanases and betaglucosidase enzymes. However, none of the bacterial strains was able to contemporaneously exhibit the highest cellulolytic activity for all the appropriate enzymes. Therefore, we presume that using bacterial pools that include the best strains for each specific enzymatic activity might be more effective. Future work will focus on the physiology of the most potential bacterial strains and experiments under controlled conditions to understand the influence of the active cellulolytic bacterial community on plant residue decomposition in soil.

**Supplementary Materials:** The following supporting information can be downloaded at: https://www.mdpi.com/article/10.3390/su15010598/s1, Table S1: Summary of molecular identification, cellulases activity, and morphological characteristics of soil bacterial isolates.

**Author Contributions:** Conceptualisation, A.S., S.S. and G.K.; methodology, A.S., S.S. and F.D.; software, A.S. and F.D.; validation, A.S., S.S., F.D. and C.V.; formal analysis, A.S.; investigation, A.S. and F.F.; data curation, A.S.; writing—original draft preparation, A.S.; writing—review and editing, A.S.; visualisation, A.S.; supervision, S.S., F.D. and C.V. All authors have read and agreed to the published version of the manuscript.

**Funding:** This research received no external funding.

**Institutional Review Board Statement:** Not applicable.

**Informed Consent Statement:** Not applicable.

**Data Availability Statement:** Not applicable.

**Conflicts of Interest:** The authors declare no conflict of interest.

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
