# Peer review of "Characterisation of Cellulolytic Bacteria Isolated from Agricultural Soil in Central Lithuania"

_sustainability, doi:10.3390/su15010598_

Round 1

Reviewer 1 Report

The authors contributed to isolating effective bacterial groups that are responsible for cellulose degradation isolated from agricultural soil. Their results are attractive and persuasive. However, I think there are still some confusions in the Material and Result parts. Please see details as follows:

#Comment 1: Line 97-99: Please add the Latin name of these crop species.

#Comment 2: Line 97-100: Did the "five-member crop rotations" practices also start from1956?

#Comment 3: Line 110-112: I am a little confused, what is "EN" mean? Please give the exact definition of that.

#Comment 4: Figure 4 and line 326: there was a "Xilosidase" in Figure 4 but a "Xylosidase" in line 326. Are they the same one? please check and change.

#Comment 5: Line 312-314 and Figure 4: How does the value "84.02171" you get? I think it is difficult to understand this figure by just using black and white color. How about trying the other colors?

#Comment 6: Line 332-333: There are actually more information in Figure 5, Please check and explain the figure more.

#Comment 7: Figure6 and line 352-356. Same questions as above (#Comment 5).

#Comment 8: Line 376-384 and Line 332-333. Why the explanation of figure 5 were departed into two parts? I think it is unreasonable to be here (here you put the main part of figure 5 information after the heat map of sugar test results). Please check the results and rearrange.

Reviewer 2 Report

Sustainability-2118486 is very interesting and give the valuable information to the researchers and readers. The subject of the manuscript is consistent with the scope of the Journal. The English language is fluent and easy to read. Thus, I suggested that the manuscript need to be minor revised before it is accepted by this journal.

1. It is suggested to supplement the effects of different temperature, pH, light and other conditions on the growth of highly efficient degrading bacteria.

2. Physiological and biochemical characteristics of highly efficient degrading strains should be supplemented.

3. Line 30: “MUF” should be full name. Please check the full text for the first time abbreviations appear.

4. Table 1 should be a three line table.

5. Line 140: Spaces should be removed.

6. The columns in Figures 4 and 6 are not clear.

7. The logic and neat of introduction need to be further improved.

8. Conclusions need to be further shorten.

Reviewer 3 Report

i have read the manuscript and contains good information and can be accepted for publication. Only my suggestion is kindly add laboratory and experiment photographs in the manuscript. 

Reviewer 4 Report

The manuscript entitled Characterisation of cellulolytic bacteria isolated from agricultural soil in central Lithuania presents interesting information on microbial communities in agricultural soils. 

in order to select better strains that participate in the degradation of cellulose in unique agroecosystems

The manuscript is well presented except for the following minor details:

Line 357: to check for the sentence and change/complete it

line 464: Suproniene? to explain/correct this term. Is this a reference? year?

line 471: conclusions: must be improved

Also, check for references and figure legends.
